# Updated Confirmatory Diagnosis for Mucopolysaccharidoses in Taiwanese Infants and the Application of Gene Variants

**DOI:** 10.3390/ijms23179979

**Published:** 2022-09-01

**Authors:** Chih-Kuang Chuang, Yuan-Rong Tu, Chung-Lin Lee, Yun-Ting Lo, Ya-Hui Chang, Mei-Ying Liu, Hsin-Yun Liu, Hsiao-Jan Chen, Shu-Min Kao, Li-Yun Wang, Huey-Jane Ho, Hsiang-Yu Lin, Shuan-Pei Lin

**Affiliations:** 1Division of Genetics and Metabolism, Department of Medical Research, MacKay Memorial Hospital, Taipei 10449, Taiwan; 2College of Medicine, Fu-Jen Catholic University, New Taipei City 24205, Taiwan; 3Department of Pediatrics, MacKay Memorial Hospital, Taipei 10449, Taiwan; 4The Rare Disease Center, MacKay Memorial Hospital, Taipei 10449, Taiwan; 5The Chinese Foundation of Health, Neonatal Screening Center, Taipei 11070, Taiwan; 6Taipei Institute of Pathology, Neonatal Screening Center, Taipei 103642, Taiwan; 7Department of Early Childhood Care and Education, MacKay Junior College of Medicine, Nursing and Management, Taipei 11260, Taiwan; 8Department of Medicine, MacKay Medical College, New Taipei City 25245, Taiwan; 9Department of Medical Research, China Medical University Hospital, China Medical University, Taichung 406040, Taiwan; 10Department of Infant and Child Care, National Taipei University of Nursing and Health Sciences, Taipei 112303, Taiwan

**Keywords:** mucopolysaccharidosis, glycosaminoglycan (GAG), autosomal recessive inheritance, X-linked recessive inheritance, variant allele, GAG-derived disaccharide

## Abstract

Mucopolysaccharidosis (MPS) is a lysosomal storage disease caused by genetic defects that result in deficiency of one specific enzyme activity, consequently impairing the stepwise degradation of glycosaminoglycans (GAGs). Except for MPS II, the other types of MPS have autosomal recessive inheritance in which two copies of an abnormal allele must be present in order for the disease to develop. In this study, we present the status of variant alleles and biochemistry results found in infants suspected of having MPS I, II, IVA, and VI. A total of 324 suspected infants, including 12 for MPS I, 223 for MPS II, 72 for MPS IVA, and 17 for MPS VI, who were referred for MPS confirmation from newborn screening centers in Taiwan, were enrolled. In all of these infants, one specific enzyme activity in dried blood spot filter paper was lower than the cut-off value in the first blood sample, as well asin a second follow-up sample. The confirmatory methods used in this study included Sanger sequencing, next-generation sequencing, leukocyte enzyme fluorometric assay, and GAG-derived disaccharides in urine using tandem mass spectrometry assays. The results showed that five, nine, and six infants had MPS I, II, and IVA, respectively, and all of them were asymptomatic. Thus, a laboratory diagnosis is extremely important to confirm the diagnosis of MPS. The other infants with identified nucleotide variations and reductions in leukocyte enzyme activities were categorized as being highly suspected cases requiring long-term and intensive follow-up examinations. In summary, the final confirmation of MPS depends on the most powerful biomarkers found in urine, i.e., the quantification of GAG-derived disaccharides including dermatan sulfate, heparan sulfate, and keratan sulfate, and analysis of genetic variants can help predict outcomes and guide treatment.

## 1. Introduction

Mucopolysaccharidosis (MPS) is a lysosomal storage disease caused by gene defects that results in deficiency of one specific enzyme activity. This deficiency blocks the stepwise degradation of glycosaminoglycans (GAGs), resulting in the accumulation of GAGs in cells, tissues, and organs, ultimately causing dysfunction of multiple organs or systems. There are seven major types of MPS: MPS I, II, III, IV, VI, VII, and IX. The pathogenesis of most of these types is due to inheritance of defective genes from parents, separately. However, MPS II is caused by X-linked recessive inheritance, and the defective gene is of maternal origin [1,2,3].

Advances in DNA technology over the past several decades have resulted in the development of enzyme replacement therapy (ERT) for MPS, which was approved by the FDA in April 2003 for MPS I (Aldurazyme^®^; laronidase), May 2005 for MPS VI (Naglazyme^®^; galsulfase), July 2006 for MPS II (Elaprase^®^; idursulfase), and February 2014 for MPS IVA (elosulfase alfa). Outcome surveys after ERT have shown satisfactory improvements in health status and reductions in disease severity, particularly when the therapy is started as early as possible [4]. However, there are some limitations, for instance, the inability of protein drugs to pass the brain–blood barrier [5], and inability to reverse skeletal deformities or dysplasia [6].Many studies have reported that the early initiation of laronidase prior to the onset of symptoms in patients with attenuated MPS I can slow or prevent the development of severe clinical manifestations [7]. In addition, another case study reported that scoliosis was found in the older affected sister of a sibling pair with MPS I at 3.6 years of age, but that scoliosis was not found in the brother who received early ERT after birth [4]. These findings show that early, possibly pre-symptomatic treatment, can improve the prognosis [5].

The nationwide newborn screening (NBS) program for MPS in Taiwan was implemented in August 2015. Except for MPS III, NBS for MPS currently includes MPS I, MPS II, MPS VI, and MPS IVA. Tandem mass spectrometry assays for MPS screening are used to quantitatively analyze alpha-L-iduronidase (IDUA), iduronate-2-sulfatase (IDS), arylsulfatase B (ARSB), and galactose-6-sulfatase (GALNS) enzyme activities in dried blood spot filter paper (DBS). The cut-off values are set at 30% of the mean activity for the first DBS and 10% of the mean activity for the second DBS. If one of the specific enzyme activities is less than the cut-off value in the first DBS, a second newborn blood spot sample is required. The false-positive rate depends on the cut-off values used for the first and second tests [8,9,10]. Infants suspected of having MPS are referred to MacKay Memorial Hospital (MMH) for a confirmatory diagnosis.

The confirmatory diagnosis of MPS is usually performed through a series of quantitative measurements, including total GAG concentrations in urine using a dye-based method (the dimethylmethylene blue [DMB]/creatinine [Cre] ratio), GAG-derived disaccharides detected by tandem mass spectrometry assay (i.e., chondroitin sulfate (CS), dermatan sulfate (DS), heparan sulfate (HS), and keratan sulfate (KS)), leukocyte enzyme fluorometric assay, and Sanger sequencing analysis to identify the defective gene. A confirmative diagnosis of MPS must meet the following criteria: (1) elevation of urinary DMB/Cre ratio; (2) extraordinarily high quantities of urinary DS, HS, or KS; (3) deficiency of one specific enzyme activity in leukocytes; and (4) the identification of nucleotide variations. A previous study reported that 48.5% of all confirmed cases (16/33) were diagnosed through the NBS program for MPS in Taiwan from 2016 to 2019 [11]. Consequently, the median age at a diagnosis of MPS has fallen dramatically, from 4.3 years to 0.2 years, since the implementation of the NBS program. In the past, most suspected cases were diagnosed due to definite clinical indications, whereas most suspected MPS patients are now referred from the NBS program for MPS with no pre-symptoms. How best to predict the prognosis of infants with MPS is especially important.

In this study, we investigated 113 variants from infants diagnosed through the NBS program for MPS I, II, IVA, and VI in Taiwan, of which 40 were classified as being pathogenic, 40 as likely pathogenic, 23 as uncertain significance, and 10 as benign, according to the guidelines published by the American College of Medical Genetics and Genomics (ACMG) [12,13]. We also present data including biochemical and molecular DNA analyses, and in vitro gene expression analysis using a COS-7 cell transfection experiment, to define the effect of a variant on the disease itself. The severity of MPS is closely related to variation pattern, i.e., missense, nonsense, small deletion, inversion, splicing, and silent mutations. A greater understanding of the characteristics of gene variants, i.e.,whether they are pathogenic or non-pathogenic, can help to predict the severity and prognosis of individual MPS types correctly, and also prompt intensive and long-term follow-up to monitor the health condition of highly suspected infants. Moreover, this will also help guide the appropriate time to initiate ERT in confirmed MPS infants.

## 2. Results

### 2.1. The Number of Referred Infants from the NBS Program and Prevalence of MPS

The nationwide NBS program for MPS was implemented in August 2015, and as of 30 April 2022, 656,896, 546,040, 170,694, and 405,935 newborn infants have been screened for MPS I, II, IVA, and VI, respectively. Of these infants, 12, 223, 72, and 17 were referred to MMH for confirmation (incidence values of 0.17, 4.08, 4.22, and 0.42/10,000 infants, respectively). Signed informed consent was obtained for 84.0%, 83.6%, 76.5%, and 80.0% of these infants, respectively. The confirmed MPS infants include five with MPS I, nine with MPS II, six with MPS IVA, and zero with MPS VI (prevalence values of 0.76, 1.64, 3.52, and 0.00/100,000 live births, respectively) (Figure 1).

### 2.2. The ACMG Classification of Gene Variants from Different MPS Types in Taiwan

The ACMG classification system is based on a rich database which can be used to differentiate sequence variants irrespective of the type of Sanger or next-generation sequencing analysis used to identify them. Variants are divided into five classes, namely benign, likely benign, uncertain significance, likely pathogenic, and pathogenic variants. Based on the ACMG classification of sequence variants, we analyzed 480 cases with or without MPS or carriers, including 36 (12 newborns) for MPS I, 328 (223 newborns) for MPS II, 94 (72 newborns) for MPS IVA, and 22 (17 newborns) for MPS VI by Sanger sequencing in order to identify the gene variants in these cases.

Apart fromthe 324 cases from the NBS program for MPS, the other suspected cases were from the outpatient department of MMH, from parents of confirmed and some highly suspected infants, as well as from other hospitals or medical centers in Taiwan for MPS confirmation. Of the three hundred and twenty-four cases referred from the NBS program, twenty were confirmed, of whom five, nine and six had MPS I, II, and IVA, respectively. MPS was excluded in 13 cases, as no variant was found. Fifty-seven cases were defined as carriers, and the rest were categorized as highly suspected MPS cases. Twelve *IDUA* variants, eleven *IDS* variants, twelve *GALNS* variants, and five *ARSB* variants were classified as pathogenic variants, of which five *IDUA* variants, seven *IDS* variants, eight *GALNS* variants, and four *ARSB* variants were from infants identified through NBS for MPS. Six *IDUA* variants, eight *IDS* variants, seventeen *GALNS* variants, and nine *ARSB* variants were classified as likely pathogenic, of which two *IDUA* variants, five *IDS* variants, fifteen *GALNS* variants, and six *ARSB* variants were from infants identified through NBS for MPS. Thirteen *IDUA* variants, five *IDS* variants, three *GALNS* variants, and two *ARSB* variants were classified as uncertain significance, of which nine *IDUA* variants, three *IDS* variants, three *GALNS* variants, and one *ARSB* variant were from infants identified through NBS for MPS. One *IDUA* variant, three *IDS* variants, three *GALNS* variants, and three *ARSB* variants were classified as benign (likely benign), of which one *IDUA* variant, three *IDS* variants, one *GALNS* variant, and three *ARSB* variants were from infants identified through NBS for MPS (Table 1, Table 2, Table 3 and Table 4). Regardless of the variant, the ACMG classification of sequence variants could be used to predict the severity and prognosis of MPS disease.

### 2.3. Number of Variants Detected in the Suspected Infants Referred from the NBS Program

In patients with X-linked recessive disorders or autosomal recessive disorders, either one or two gene variants should be found by Sanger sequencing, respectively. However, the absence of one or more pathogenic mutations is common, particularly in patients with autosomal recessive disorders such as MPS I, IVA, and VI. DNA molecular analysis of the infants suspected of having MPS was used to identify the number of variants, and the incidence rates from zero to three (or four) variants were calculated (Figure 2A,B).

#### 2.3.1. Results Found in MPS I

For MPS I, two infants did not have any variants, thereby ruling out the possibility of having MPS I, and the false-positive referral rate from NBS was 16.7%. One variant was found in an infant who was defined as a carrier (8.3%). Two variants were found in seven infants, of whom five were confirmed to have MPS I. The other two infants were defined as being highly suspected of having MPS I, and they both had reductions in IDUA enzyme activity (1.2 and 2.1 µmol/g protein/h vs. reference value 5.9–27.8 µmol/g protein/h) and normal DS and HS quantification in urine (<0.80 µg/mL for DS and <0.78 µg/mL for HS). Two infants had three variants with reductions in IDUA enzyme activity (1.5 and 0.95 µmol/g protein/h) and normal DS and HS quantification in urine.

#### 2.3.2. Results Found in MPS II

Of the 223 infants suspected of having MPS II, one did not have any variants, thereby ruling out the possibility of having MPS II, and the false-positive referral rate from NBS was 0.90%. One variant was found in sixty-twoinfants (27.8%), of whom nine were confirmed to have MPS II. The other 53 infants were defined as being highly suspected of having MPS II (23.14%), in whom variants were identified. These infants had normal or reduced leukocyte IDS enzyme activity, and negative urinary GAG-derived disaccharide (DS and HS) quantification. These cases were required to receive long-term and intensive follow-up examinations in order to control the progressive health condition. Remarkably, 71.7% (159/223) of the referred infants had a combination of the following four mutants: c.103+34_56dup [−] + c.684A>G [p.Pro228=] + c.851C>T [p.P284L] + c.1180+184T>C [−]. Although these infants had four mutants, they showed reductions in IDS enzyme activity (5.5 ± 4.8µmol/g protein/4h vs. reference value 12.89–131.83 µmol/g protein/4 h), normal DS and HS quantification in urine, and no symptoms.

#### 2.3.3. Results Found in MPS IVA

A total of 72 infants were referred to MMH for MPS IVA confirmation. No mutants were found in eight of these infants, and the false-positive referral rate from NBS was 11.1% (8/72). Forty-five infants had only one variant, thus, 62.5% (45/72) of the infants were carriers of MPS IVA. The GALNS enzyme activity in this carrier group was 2.3 ± 1.09 µmol/g protein/h (reference value 5.9–27.8 µmol/g protein/h), and the urinary KS quantification by tandem mass spectrometry was lower than the cut-off value (7.9 µg/mL). Nineteen infants had two mutants in the *GALNS* gene. Of these infants, six without MPS symptoms were confirmed according to the deficiency of leukocyte GALNS enzyme activity (0.62 ± 0.39 µmol/g protein/h) and increased quantity of urinary KS (21.90 ± 13.68 µg/mL; reference value < 7.9 µg/mL). The other 13 infants were categorized as the highly suspected group, in which the GALNS enzyme activity was 1.51 ± 0.87 µmol/g protein/h, but the urinary KS quantities were lower than the cut-off value.

#### 2.3.4. Results Found in MPS VI

For MPS VI, two infants did not have any variants, thereby ruling out the possibility of having MPS VI, and the false-positive referral rate from NBS was 11.8%. Eleven infants had one mutation, and they were categorized as carriers. In these infants, the ARSB enzyme activity was slightly lower than the minimum reference value (13.53 ± 7.39 µmol/g protein/h vs. reference value 14–228 µmol/g protein/h); in addition, the urinary DS quantification was less than the cut-off value. Three infants had two mutants, and they were classified as being highly suspected of having MPS VI. The ARSB enzyme activity (12.03 ± 4.52 µmol/g protein/h) was slightly lower than the reference value, and the DS quantities detected by tandem mass spectrometry assay were all less than the cut-off value. No confirmed MPS VI infants have been found since the NBS program was implemented in August 2015.

### 2.4. Variations of MPS Genes According to ACMG Classification

#### 2.4.1. The ACMG Classification of Variants Found in Infants with Confirmed and Suspected MPS I

Five infants were confirmed to have MPS I, including a male and female sibling, a twin sister, and an individual infant with no relationship to the former two families. The cDNA change in the sibling was c.300-3C>G [−]+ c.1874A>C [p.Y625S], c.1037T>G [p.L346R] + c.1091C>T [p.364M] in the twin sister, and c.300-3C>G [−] + c.1395delC [p.G466AfsTer59] in the individual infant. According to ACMG classification, both c.1037T>G [p.L346R] and c.1395delC [p.G466AfsTer59] were classified as pathogenic variants; c.300-3C>G [−] and c.1874A>C [p.Y625S] were classified as likely pathogenic variants; and c.1091C>T was classified as uncertain significance.

#### 2.4.2. The ACMG Classification of Variants Found in Infants with Confirmed and Suspected MPS II

Nine infants with no family relationship were confirmed to have MPS II. The nucleotide variations c.254C>T [p.A85V], c.311A>T [p.D104V], c.1025A>G [p.H342R], c.1400C>T [p.P467L], c.1007-1666_c.1180+2113delinsTT [−], and *IDS* inversion [−] were classified as pathogenic variants, and caused a deficiency in leukocyte IDS enzyme activity ranging from 0.27 to 0.99 µmol/g protein/4h. Two infants had a point mutation c.817C>T [p.R273W], which was classified as an uncertain significance variant, and caused deficiencies in IDS enzyme activity of 0.2 and 0.4 µmol/g protein/4h, respectively. Of note, the urinary DS and HS quantities obtained from these nine confirmed MPS II infants were higher, ranging from 5.48 to 30.77 µg/mL for DS, and from 1.83 to 203.35 µg/mL for HS (cut-off values: <0.80 for DS and <0.78 µg/mL for HS).

The other nucleotide variations c.805G>A [p.D269N], c.890G>A [p.R297H], and c.589C>T [p.P197S] were classified as pathogenic variants, and caused a reduction in leukocyte IDS enzyme activity ranging from 7.80 to 17.68 µmol/g protein/4 h. Five variants, c.1478G>A [p.R493H] (*n* = 8), c.1513T>C [p.F505L], c.659T>C [p.F220S], c.851C>T [p.P284L] (*n* = 4), and c.142C>T [p.R48C] (*n* = 2) were classified as likely pathogenic variants, and caused a reduction in (or did not affect) leukocyte IDS enzyme activity ranging from 5.93 to 23.9 µmol/g protein/4 h. Five suspected MPS II infants had the variant c.301C>T [p.R101C], which was classified as an uncertain significance variant and did not affect leukocyte IDS enzyme activity (26.1 ± 9.9 µmol/g protein/4 h). Twenty-two infants carried variation c.1499C>T [p.T500I], which was classified as a benign variant and did not affect leukocyte IDS enzyme activity (26.1 ± 9.9 µmol/g protein/4 h). Of note, urinary DS and HS quantities obtained from these infants were all normal.

#### 2.4.3. The ACMG Classification of Variants Found in Infants with Confirmed or Suspected MPS IVA

Six infants with no family relationship were confirmed to have MPS IVA, including nucleotide variations c.953T>G [p.M318R]+[−],c.857C>T [p.T286M]+c.953T>G [p.M318R], c.638C>T [p.A213V]+c.953T>G [p.M318R], c.190_191delinsAT [p.A64I] +c.1108C>T [p.P370S], c.953T>G [p.M318R]+c.1108C>T [p.P370S], and c.808G>T [p.Glu270Ter]+c.857C>T [p.T286M]. In these infants, c.953T>G [p.M318R] and c.857C>T [p.T286M] were classified as pathogenic variants, and caused a deficiency in leukocyte GALNS enzyme activity ranging from 0.00 to 1.00 µmol/g protein/h. Five of the six confirmed MPS IVA infants carried either c.953T>G [p.M318R] or c.857C>T [p.T286M], which are hot spots found in about 23.8% and 28.6% of Taiwanese infants with Morquio syndrome, respectively [10]. Nucleotide variationsc.190_191delinsAT [p.A64I], c.638C>T [p.A213V], c.808G>T [p.Glu270Ter], and c.1108C>T [p.P370S] were classified as likely pathogenic with a 90% chance of pathogenicity, and they have been reported to cause a major problem due to the protein produced [14]. Eight infants had heterogeneity of two variants, including c.887C>T [p.A296V]+c.953T>G [p.M318R] (*n* = 2), c.857C>T [p.T286M]+c.953T>G [p.M318R] (*n* = 2), c.857C>T [p.T286M]+c.1127G>A [p.R376Q], c.857C>T [p.T286M]+c.*3C>G [−], c.131G>T [p.G44V]+c.985C>A [p.H329N], c.704C>A [p.T235K]+c.887C>T [p.A296V]. Of these variants, c.131G>T [p.G44V] was classified as pathogenic, c.704C>A [p.T235K] and c.985C>A [p.H329N] as likely pathogenic, and c.887C>T [p.A296V] and c.*3C>G [−] as uncertain significance. The leukocyte GALNS enzyme activity was mostly reduced in these infants, ranging from 0.5 to 3.5 µmol/g protein/h, and the urinary KS quantities detected by tandem mass spectrometry were all less than the cut-off value. These eight infants were defined as being highly suspected of having MPS IVA, and they were required to receive long-term and intensive follow-up examinations and clinical inspection. The other infants were carriers who carried one variant causing a significant decrease of leukocyte GALNS enzyme activity, ranging from 1.5 to 4.4 µmol/g protein/h, and negative urinary KS quantification.

#### 2.4.4. The ACMG Classification of Variants Found in Infants with Suspected MPS VI

No confirmed MPS VI infants have been identified through the NBS program in Taiwan. Two infants had heterogeneity of two variants, c.424A>G [p.M142V] + c.1072G>A [p.V358 M], and c.43C>G [p.P15A] + c.245T>C [p.L82P]. Of these variants, c.424A>G [p.M142V] and c.245T>C [p.L82P] were classified as likely pathogenic, c.43C>G [p.P15A] as uncertain significance, and c.1072G>A [p.V358 M] as benign. The leukocyte ARSB enzyme activity was normal in these infants (13.6 and 17.5 µmol/g protein/h, respectively; reference value14–228µmol/g protein/h). In addition, the urinary DS quantities were all less than the cut-off value (<0.80 µg/mL). Unusually, one infant had three variants, c.313-26T>C [−] + c.1143-27A>C [−] + c.1072G>A [p.V358 M], all of which were classified as benign and led to lower leukocyte ARSB enzyme activity than normal (approximately 5.0 µmol/g protein/h). Urinary DS quantification was normal in these infants. The other11 infants were carriers, and the leukocyte ARSB activity was relatively lower or slightly higher thanthe minimum of the reference value. Of note, one infant carried the variant c.478C>T [p.R160Ter], which caused very low ARSB enzyme activity (1.0 µmol/g protein/h); however, the urinary DS quantification was normal.

### 2.5. Expression of Enzyme Activity by In Vitro COS-7 Cell Transfection Assay

Mutant gene expressions of enzyme activity were examined using an in vitro COS-7 cell transfection assay to evaluate the effect of one specific mutant allele. Six, 16, and 12 novel genes were identified in infants from the NBS program with MPS I, II, and IVA, respectively.

(1) For MPS I (*IDUA* gene), six novel mutants were analyzed, including c.2T>C [p.M1T], c.343G>A [p.D115N], c.355G>T [p.D119Y], c.1081G>A [p.D119Y], c.1816G>T [p.V606L], and c.1874A>C [p.Y625S], in which the percentages of wild-type IDUA enzyme activities were 84.3%, 12.4%, 13.7%, 100.0%, 40.3%, and 0.0%, respectively.

(2) For MPS II (*IDS* gene), a total of 16 novel mutants were identified, of which 12 and the IDS activity expressed in percentage of wild-type have been reported previously, including c.142C>T [p.R48C], c.254C>t [p.A85V], c.311A>T [p.D104V], c.589C>t [p.P197S], c.778C>t [p.P260S], c.817C>T [p.R273W], c.851C>T [p.P284L], c.890G>A [p.R297H], c.1025A>G [p.H342R], c.1400C>T [p.P467L], c.1478G>A [p.R493H], and c.1499C>T [p.T500I] [15]. The other four novel IDS genes were c.659T>C [p.F220S], c.684A>G [p.Pro228=], c.805G>A [p.D269N], and c.1513T>C [p.F505L], and the percentages of IDS activity expressed in percentage of wild-type were 0.0%, 100.0%, 0.0%, and 84.58%, respectively.

(3) For MPS IVA, there were 12 novel mutants in the *GALNS* gene, including c.131G>T [p.G44V], c.190_191delinsAT [p.A64I], c.374C>T [p.P125L], c.638C>T [p.A213V], c.706C>T [p.H236Y], c.782T>C [p.I261T], c.857C>T [p.T286M], c.887C>T [p.A296V], c.953T>G [p.M318R], c.985C>A [p.H329N], c.1108C>T [p.P370S], and c.1496C>T [p.P499L]. The effect of these mutations on GALNS activity was investigated by transfecting the mutant plasmids into COS-7 cells, human 293T cell line, and human MPS IVA fibroblast cell line (purchased from Coriell Institute for Medical Research, Camden, NJ, USA) for expression studies. However, the GALNS expression experiments failed, and no differences were found between the mutants and wild-type.

### 2.6. Quantification of Urinary GAG-Derived Disaccharides by Tandem Mass Spectrometry Assay Was Crucial to Confirm the Diagnosis of MPS

Results of the DMB/Cre ratio and GAG-derived disaccharides (DS, HS, and KS) for the pseudo-deficiency of MPS in the highly suspected and confirmed infants are shown below. The reference value for the DMB/Cre ratio in the infants’ age group (<6 months) was <70.68 mg/mmol creatinine (mean ± 2SD = 41.83 ± 28.85); the DS and HS cut-off values were <0.80 and <0.78 μg/mL, respectively.

For MPS I, the average DMB/Cre ratio was 29.31 ± 5.85 mg/mmol creatinine, and the DS, HS, and KS quantities were 0.11 ± 0.08, 0.13 ± 0.08, and 2.14 ± 2.26 µg/mL for the pseudo-deficiency of MPS I in the highly suspected infants (*n =* 7), compared to 117.2 ± 81.74 mg/mmol creatinine, 104.57 ± 104.34, 40.26 ± 40.24, and 1.14 ± 0.74 µg/mL in the confirmed MPS I infants (*n =* 5), respectively.

For MPS II, the average DMB/Cre ratio was 39.87 ± 11.25 mg/mmol creatinine, and the DS, HS, and KS quantities were 0.14 ± 0.09, 0.27 ± 0.20, and 1.58 ± 1.18 µg/mL for the pseudo-deficiency of MPS II in the highly suspected infants (*n* = 214), compared to 98.1 ± 39.23 mg/mmol creatinine, 14.06 ± 8.25, 63.50 ± 69.57, and 1.88 ± 2.29 µg/mL in the confirmed MPS II infants (*n =* 9), respectively.

For MPS IVA, the average DMB/Cre ratio was 31.25 ± 11.07 mg/mmol creatinine, and the DS, HS, and KS quantities were 0.15 ± 0.13, 0.14 ± 0.13, and 1.56 ± 1.90 µg/mL for the pseudo-deficiency of MPS IVA in the highly suspected infants (*n =* 66), compared to 62.02 ± 25.39 mg/mmol creatinine, 0.06 ± 0.06, 0.03 ± 0.02, and 20.72 ± 12.81 µg/mL in the confirmed MPS IVA infants (*n =* 6), respectively.

For MPS VI, the average DMB/Cre ratio was 28.21 ± 11.17 mg/mmol creatinine, and the DS, HS, and KS quantities were 0.27 ± 0.19, 0.12 ± 0.09, and 0.76 ± 0.59 µg/mL for the pseudo-deficiency of MPS VI in the highly suspected infants (*n =* 17), however no confirmed MPS VI infants were found.

## 3. Discussion

Since the NBS program for MPS was implemented in Taiwan, more than 700,000 infants have been screened, of whom 324 have been referred to MMH for a confirmatory diagnosis. In this study, many highly suspected MPS infants with lower enzyme activity had variants defined as pathogenic or likely pathogenic according to ACMG classification. These cases were not diagnosed with MPS due to the normal quantification of GAG-derived disaccharides measured by MS/MS-based assay.

These findings raise the issue of how to distinguish between confirmed and highly suspected MPS infants with pseudo-deficiency mutations. The most important biomarkers used to confirm the diagnosis of MPS are individual GAG-derived disaccharides in urine, i.e., DS, HS, or KS. In general, abnormally increased quantities of DS and HS were found in the urine of the confirmed MPS I and MPS II infants. However, only an increase of DS was found in the confirmed MPS VI infants, and only an increase of KS was found in the confirmed MPS IVA infants. Validation of MS/MS-based assays for GAG-derived disaccharides showed comparatively good precision and accuracy. The average within-run and between-run precisions of DS, HS, and KS assays were good, and the coefficients of variance were all less than 10%. In addition, the recovery of the MS/MS assay was acceptable (94.2%), and the linearity of linear regression showed very good accuracy (correlation coefficient (*r*) 0.9992) [16]. In this study, the quantities of DS, HS, and KS were calculated using the CS-normalized method instead of the conventional creatinine-normalized method. Using this method, the sensitivity (true positive rate), specificity (true negative rate), and positive predictive value were excellent, and could effectively decrease the occurrence of false-positive or false-negative results. The accuracy rates of the CS-normalized method were 100% for DS, 95.5% for HS, and 100% for KS, compared to 79.1% for DS, 80.6% for HS, and 79.1% for KS, when using the creatinine-normalized method [17].

ACMG classification of gene variants is valuable and can be used to predict disease severity, features, and prognosis (genotyping). ACMG classification integrates individual variants of particular genes into a large database that contains related information or evidence either of clinical manifestations (phenotyping) or from laboratory experiments. The consistency of correlations between genotyping and phenotyping is established mainly for the evaluation of disease prognosis, which is very important when planning the future care of patients. In this study, we presented all variations of the MPS genes in the infants referred from the NBS program for MPS, including the *IDUA*, *IDS*, *GALNS*, and *ARSB* genes. In this study, we did not perform cis vs. trans analysis or analyze multiple mutations in a single allele; however, we performed Sanger sequencing for each confirmed and some of the highly suspected MPS infants and their parents for MPS I, IVA, and VI, and maternal DNA analysis was required mainly for the MPS II infants. In this way, we could precisely identify whether the variants were inheritedfrom the father or mother.

A pathogenic variant would lead to deficiency of one specific enzyme activity, which may ultimately result in particular signs and symptoms of MPS. Furthermore, the likelihood of developing these signs and symptoms would be higher than in those with likely pathogenic, uncertain significance, or benign variants. Therefore, two categories of subjects can be defined: confirmed MPS infants, and highly suspected MPS infants. Asymptomatic infants in both groups are required to undergo intensive long-term clinical management and laboratory examinations so that ERT can be initiated in a timely fashion. The standard operating procedures of this intensive follow-up management are presented in Figure 3.

For the confirmed and highly suspected infants, our genetic counselor recalls and arranges outpatient department visits at our hospital every 6 months for 3 years of follow-up. During long-term follow-up, our pediatric geneticists and genetic counselor pay particular attention to the parents’ concerns, maintaining a caring and enthusiastic approach to ease their anxiety. The overall objectives of the NBS program and the subsequent diagnostic procedures are as follows: (1). Plan a 6-month recall process depending on the judgment of a pediatric geneticist and based on individual cases. (2). Recall asymptomatic MPS infants back to MMH for MPS follow-up examinations, including regular physical examinations for the earliest presenting symptoms such as otitis media, abdominal or inguinal hernia, gibbous, and coarse facial features, as well as urinary biochemistry GAG tests. (3). Give ERT before the typical signs or symptoms of MPS develop. (4). Define the attenuated form and pseudo-deficiency of MPS. (5). Achieve the aim of NBS for MPS: “Early detection, making an early diagnosis, and providing early therapy can effectively prevent the development of severe clinical manifestations.”

## 4. Materials and Methods

The diagnosis of MPS was confirmed using biochemical and molecular analyses, including urinary total GAG quantitative measurements, GAG-derived disaccharide quantification using an MS/MS-based method, leukocyte enzyme assay, and variations of one assigned MPS gene by Sanger sequencing. In addition, RNA sequencing was also used in cases where one of the definitive variations was unavailable.

### 4.1. Suspected MPS Infants Referred to MMH for Confirmation

A total of 324 suspected infants including 12 for MPS I, 223 for MPS II, 72 for MPS IVA, and 17 for MPS VI, who were referred from two newborn screening centers (The Chinese Foundation of Health and Taipei Institute of Pathology, Neonatal Screening Centers of Taiwan) for MPS confirmation were enrolled. Written informed consent (IRB#: 16MMHIS153 issued by MMH) was obtained from all participants and/or their legal guardian/s. Urine samples were stored at −20 °C prior to GAG analyses, and blood samples were kept at room temperature and 4 °C before leukocyte isolation for enzyme assay and molecular DNA analysis, respectively.

### 4.2. Quantification of GAGs; the DMB/Cre Ratio

The concentrations of GAGs in urine were determined quantitatively using the dye DMB, which does not require prior precipitation of the GAGs. The color was measured immediately at a wavelength of 520 nm. The DMB/Cre ratio was calculated by dividing urine creatinine by the GAG volume in mg/L, and expressed as mg/mmol creatinine [18]. The DMB/Cre ratio gives an estimation of the GAG concentration in urine and it is age-dependent, meaning that the lower the age, the higher the DMB/Cre ratio. The DMB/Cre ratio can be used to diagnose MPS, but it cannot be used to determine the type of MPS.

### 4.3. GAG-Derived Disaccharide Quantification by LC-MS/MS-Based Assay

Liquid chromatography-tandem mass spectrometry (LC-MS/MS)-based assays were used for GAG-derived disaccharide quantification (i.e., CS, DS, HS, and KS) as reported previously [16,19,20]. The LC-MS/MS-based assay was performed on a 4000 QTRAP LC-MS/MS system (AB Sciex, Foster City, CA, USA) equipped with a TurboIonSpray (electrospray ionization; ESI, Hong Kong, China), and Agilent 1260 Infinity HPLC pump and autosampler (Agilent Technologies, Santa Clara, CA, USA). A multiple reaction monitoring experiment giving a precursor ion (Q1) and a respective product ion (Q3) was applied. The instrument was operated in a positive-ion [M+H]^+^ mode. Calibrations of GAG standards and internal standards were performed with every batch of samples.

Methanolysis (chemical hydrolysis) and keratanase II (specific enzymatic digestion) were used for the derivative of individual GAG-derived disaccharides prior to the tandem mass spectrometry. The internal standards including [^2^H_6_] DS, [^2^H_6_] CS, and [^2^H_6_] HS, were derived in-house from deuteriomethanolysis of GAG standards comprising DS, CS, and HS, according to the method reported by Zhang et al. [21]. The *m/z* (mass to charge) of the parent ion and its daughter ion after collision was 426.1→236.2 for DS, 384.2→161.9 for HS, 462.0→97.0 for KS. The *m/z* of chondrosine (GlycoSyn; Lower Huf, New Zealand), which was used as an internal standard in this assay, was 353.9→73.0. We used an empirical method to calculate levels of GAG-derived disaccharides based on the quantity (peak areas) of CS with the aim of making the diagnosis of MPS more accurate and reducing the occurrence of either false positive or false negative results [17]. The overall validation of the LC-MS/MS assay for GAG-derived disaccharides was excellent. The intra-assay and inter-assay precisions (coefficients of variance) were both less than 10% (7.38% and 8.67%, respectively). The recovery of the tandem mass spectrometry assay was acceptable (94.2%), and the linearity of linear regression showed very good accuracy, with a correlation coefficient (*r*) of 0.9992.

### 4.4. Leukocyte Enzyme Activity Assay

The assay for individual enzyme activity was performed using 4-methylumbelliferyl (4-MU) substrate, i.e., 4-MU-α-L-iduronide, 4-MU-α-L-iduronide-2-sulfate, and 4-MU-beta-d-galactopyranoside-6-sulfate for MPS I, II and IVA measurements, respectively. Enzyme activity was proportional to the amount of liberated fluorescence detected (μmol enzyme activity/g protein/h). Individual enzyme activity thatwas 5% lower than normal was defined as a marked reduction (deficiency) in that enzyme activity. The reference values for enzyme activities were 4.87~54.7 μmol/g protein/h for IDUA, 12.89~131.83 μmol/g protein/4 h for IDS, and 5.9~27.8 μmol/g protein/h for GALNS [8]. We used the chromogenic substrate p-nitrocatechol sulfate to measure ARSB enzyme activity (for MPS VI) using a colorimetric method, in which the absorbance detected by a spectrophotometer at a wavelength of 515 nm was proportional to the amount of enzyme activity [22]. The reference value for ARSB enzyme activity was 14~228 μmol/gprotein/h.

### 4.5. Molecular DNA Analysis

DNA was extracted from EDTA blood. The primers were designed, and the melting temperatures (Tm) of the primers for PCR experiments were determined. The PCR product was authorized for DNA sequencing analysis by a qualified bio-technology company (Genomics) in Taipei (ISO/IEL 17025). Data analysis and aligned read sequence of data were performed to determine the variation alleles. Finally, we reviewed the literature and ACMG database to classify the gene variants (i.e., cDNA change and amino acid change) as pathogenic, likely pathogenic, uncertain significance, and benign [12,13].

### 4.6. RNA Sequencing Analysis

According to previous studies, except for MPS II, about 14% of enzymatically confirmed MPS patients (alleles) lack one or more pathogenic mutations [23]. RNA sequencing can be used to detect the mutation alleles thatcannot be identified by Sanger sequencing or next-generation sequencing. In the experiment, enrichment of the reverse transcription was performed first in order to transcribe whole mRNA in the cells to cDNA. This was then used for sequencing analysis to rapidly detect the mRNA expression.

### 4.7. Expression of Enzyme Activity by In Vitro COS-7 Cell Transfection Assay

The protocol of the in vitro COS-7 transfection assay has been reported previously [15]. COS-7 cells were cultured in Dulbecco’s MEM (Gibco; Thermo Fisher Scientific, Grand Island, NJ, USA) supplemented with 10% fetal bovine serum (Gibco; Thermo Fisher Scientific, Grand Island, NJ, USA) and penicillin-streptomycin mixture (Gibco; Thermo Fisher Scientific, Grand Island, NJ, USA) at 37 °C in 5% CO_2_. The COS-7 cells were transfected by pCMV6-Entry plasmids of wild-type specific enzyme cDNA and the mutants using Lipofectamine 3000 (Invitrogen, Carlsbad, CA, USA) following the manufacture’s protocol. After 24 h of incubation at 37 °C, the cells were harvested for specific enzyme assays. These experiments were performed in triplicate.

## 5. Conclusions

Almost all new MPS patients are referred from one of three newborn screening centers in Taiwan. A confirmatory diagnosis is made according to the positive quantification of urinary GAG-derived disaccharides, deficiency in leukocyte enzyme activity assay, and verification of MPS gene mutations. Of these methods, increased quantities of urinary DS, HS, or KS are the most important, and the results of genetic variant analysis can help predict the outcomes and guide treatment. Highly suspected MPS infants, particularly those suspected of having an attenuated or mild form, should receive long-term and intensive follow-up examinations due to the late onset of MPS manifestations. Initiating ERT in a timely fashion is very important, as it can effectively reduce or preventthe occurrence of irreversible signs or symptoms.

## Figures and Tables

**Figure 1 ijms-23-09979-f001:**
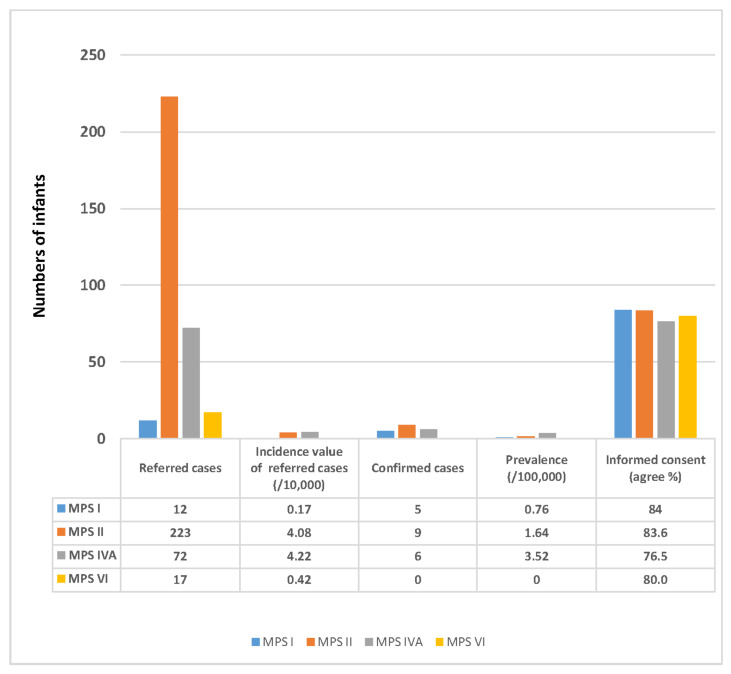
The number of infants referred for each type of MPS from the newborn screening program and the prevalence of each type of MPS.The first column is the number of referred cases; the second column shows the incidence (/10,000) of referred cases; the third column shows the number of confirmed MPS cases by MPS type; the four column shows the prevalence of individual MPS types; and the fifth column shows the percentages of the guardians who signed inform consent by MPS type.

**Figure 2 ijms-23-09979-f002:**
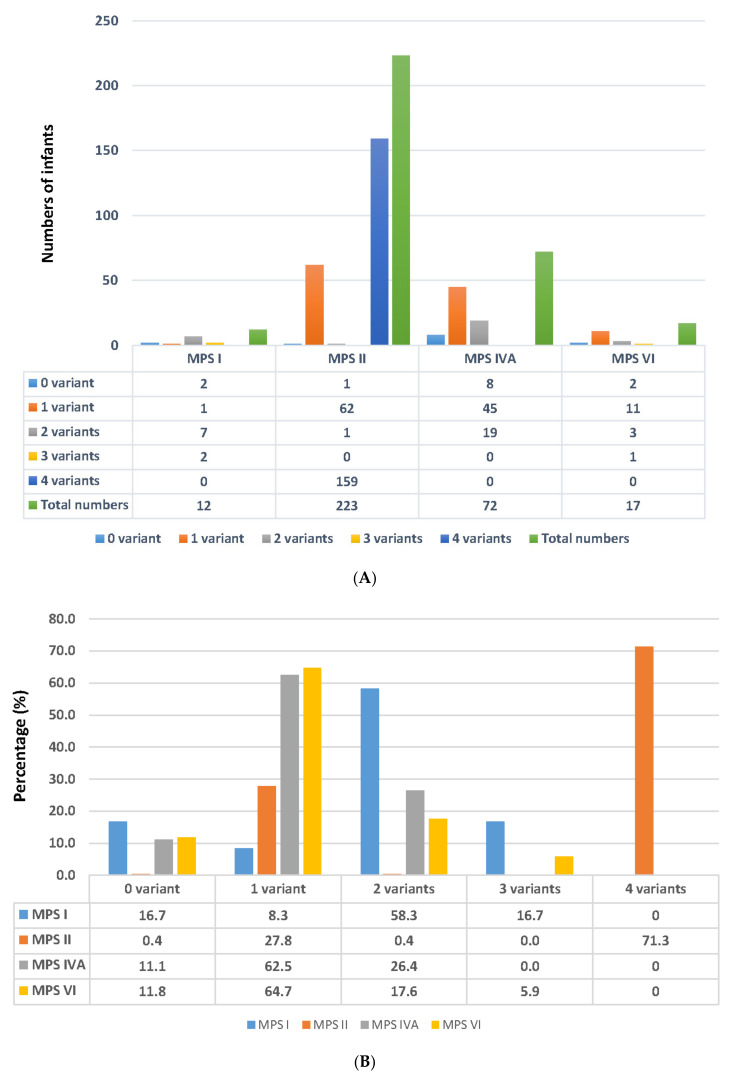
The number of variants detected in the suspected referred infants.The number of variants in the suspected cases by type of MPS (**A**); the incidence ratesof MPS according to the number of variants (**B**). The number of cases with zerovariants indicates the false-positive rate (%) of referred cases from newborn screening centers.

**Figure 3 ijms-23-09979-f003:**
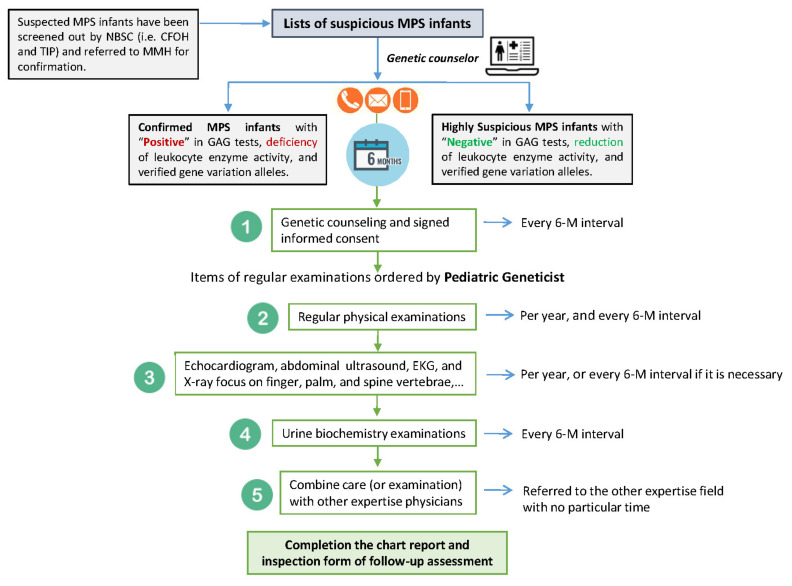
The standard operation procedures of the intensive follow-up management protocol for the infants with confirmed MPS and those highly suspected of having MPS.

**Table 1 ijms-23-09979-t001:** Pathogenic variants of MPS genes, i.e., *IDUA*, *IDS*, *GALNS*, and *ARSB* defined by the ACMG Classification.

Pathogenic Variants					
**Gene**	**cDNA change**	**Amino acid change**	**Gene**	**cDNA change**	**Amino acid change**
** *IDUA* ** ** (MPS I) **	c.2T>C	p.M1T	** *IDS* ** ** (MPS II) **	c.137A>C	p.D46A
	c.265C>T	p.R89W		c.253G>A	p.A85T
	c.532G>A	p.E178K		c.254C>T	p.A85V
	c.590G>T	p.G197V		c.311 A>T	p.D104V
	c.606C>G	p.Y202Ter		c.589C>T	p.P197S
	c.617C>T	p.S206L		c.797C>G	p.P266R
	c.911delT	p.V304GfsTer13		c.805G>A	p.D269N
	c.1037T>G	p.L346R		c.890G>A	p.R297H
	c.1395delC	p.G466AfsTer59		c.998C>T	p.S333L
	c.1422_1423dupCT	p.Y475SfsTer51		c.1025A>G	p.H342R
	c.1861C>T	p.R621Ter		c.1400C>T	p.P467L
	c.1898C>T	p.S633L			
**Gene**	**cDNA change**	**Amino acid change**	**Gene**	**cDNA change**	**Amino acid change**
** *GALNS* ** ** (MPS IVA) **	c.131G>T	p.G44V	** *ARSB* ** ** (MPS VI) **	c.478C>T	p.R160Ter
	c.139G>A	p.G47R		c.943C>T	p.R315Ter
	c.169C>T	p.P57S		c.1197C>G	p.F399L
	c.265G>T	p.G89Ter		c.1350G>T	p.W450C
	c.281G>A	p.R94H		c.1394C>G	p.S465Ter
	c.319G>A	p.A107T			
	c.374C>T	p.P125L			
	c.857C>T	p.T286M			
	c.953T>G	p.M318R			
	c.1009delC	p.H337TfsTer19			
	c.1019G>A	p.G340D			
	c.1483-2A>G				

**Table 2 ijms-23-09979-t002:** Likely pathogenic variants of MPS genes, i.e., *IDUA*, *IDS*, *GALNS*, and *ARSB* defined by the ACMG Classification.

Likely Pathogenic					
**Gene**	**cDNA change**	**Amino acid change**	**Gene**	**cDNA change**	**Amino acid change**
** *IDUA* ** ** (MPS I) **	c.300-3C>G		** *IDS* ** ** (MPS II) **	c.142C>T	p.R48C
	c.590-2A>C			c.659T>C	p.F220S
	c.1139A>G	p.Q380R		c.851C>T	p.P284L
	c.1359_1384del26bp	p.S453RfsTer47		c.880-2A>T	
	c.1634delAinsGGG	p.E545GfsTer16		c.1106C>G	
	c.1874A>C	p.Y625S		c.1188delA	p.Q396HfsTer44
				c.1478G>A	p.R493H
				c.1513T>C	p.F505L
**Gene**	**cDNA change**	**Amino acid change**	**Gene**	**cDNA change**	**Amino acid change**
** *GALNS* ** **(MPS IVA)**	c.106_111delCTGCTC	p. L36_Lu37del	** *ARSB* ** **(MPS VI)**	c.215T>C	p.L72P
	c.190_191delinsAT	p.A64I		c.245T>C	p.L82P
	c.226A>C	p.Asn76His		c.395T>C	p.L132P
	c.430G>A	p.Gly144Ser		c.424A>G	p.M142V
	c.265G>C	p.Gly89Arg		c.716A>G	p.Q239R
	c.641T>C	p.Leu214Pro		c.905G>A	p.G302E
	c.425A>G	p.H142R		c.908G>A	p.G303E
	c.638C>T	p.A213V		c.1033C>T	p.R345W
	c.704C>A	p.T235K		c.1277A>G	p.N426S
	c.706C>T	p.H236Y			
	c.782T>C	p.I261T			
	c.971C>A	p.A324E			
	c.985C>A	p.H329N			
	c.1108C>T	p.P370S			
	c.1127G>A	p.R376Q			
	c.1493C>T	p.P498L			
	c.1496C>T	p.P499L			
	c.1498G>T	p.G500C			

**Table 3 ijms-23-09979-t003:** Uncertain significance variants of MPS genes, i.e., *IDUA*, *IDS*, *GALNS*, and *ARSB* defined by the ACMG Classification.

Uncertain Significance					
**Gene**	**cDNA change**	**Amino acid change**	**Gene**	**cDNA change**	**Amino acid change**
** *IDUA* ** **(MPS I)**	c.76G>A	p.A26T	** *IDS* ** **(MPS II)**	c.103+34_56dup	
	c.95T>G	p.V32G		c.301C>T	p.R101C
	c.179A>C	p.Q60P		c.817C>T	p.R273W
	c.343G>A	p.D115N		c.1006+5G>C	
	c.355G>T	p.D119Y		c.1122C>T	p.Gly374=
	c.590-7G>C				
	c.1079T>G	p.F360C	**Gene**	**cDNA change**	**Amino acid change**
	c.1091C>T	p.T364M	** *ARSB* ** **(MPS VI)**	c.43C>G	p.P15A
	c.1093C>G	p.L365V		c.113_121del9	p.G38_G40del3
	c.1195_1197delGAG	p.E399del			
	c.1463G>C	p.R488P			
	c.1816G>T	p.V606L			
	c.1828+5G>A				
**Gene**	**cDNA change**	**Amino acid change**			
** *GALNS* ** **(MPS IVA)**	c.887C>T	p.A296V			
	c.1567T>G	p.Ter523Eext*92			
	c.*3C>G				

**Table 4 ijms-23-09979-t004:** Benign (likely benign) variants of MPS genes, i.e., *IDUA*, *IDS*, *GALNS*, and *ARSB* defined by the ACMG Classification.

Benign					
**Gene**	**cDNA change**	**Amino acid change**	**Gene**	**cDNA change**	**Amino acid change**
** *IDUA* **	c.1081G>A	p.A361T	** *IDS* **	c.684A>G	p.Pro228=
				c.1180+184T>C	
				c.1499C>T	p.T500I
**Gene**	**cDNA change**	**Amino acid change**	**Gene**	**cDNA change**	**Amino acid change**
** *GALNS* **	c.121-210C>T		** *ARSB* **	c.313-26T>C	
	c.345C>T	p.Gly115=		c.1072G>A	p.V358 M
	c.692C>G	p.Ala231Gly		c.1143-27A>C	

## Data Availability

Not applicable.

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
