# Peer review of "Updated Confirmatory Diagnosis for Mucopolysaccharidoses in Taiwanese Infants and the Application of Gene Variants"

_ijms, 2022, doi:10.3390/ijms23179979_

Round 1

Reviewer 1 Report

The authors have submitted a very important follow-up (first since 2018) of the newborn screening program for many MPS diagnoses (MPS I, II, IVa, VI) in Taiwan.  This is extremely informative for clinicians i that should be followed in evaluating infants identified through newborn screening.   

Suggestions/Questions

Abstract:  "cut-off value in a continuous second blood sample"--not sure what continuous refers to.

Introduction: "fallen dramatically from 4.3 to 0.2 years"--unclear how the number is so low if only 50% of infants participate in the newborn screening program

2.2.  Results:  "480 cases with or without MPS or carriers"--but when we add up the case numbers it total around 480, not sure where the numbers without MPS or carriers are counted?

2.3.4 Results:  "Three infants had two mutants..."--highly suspected but near normal ASB enzyme and normal GAG.  Did the authors consider cis versus trans or multiple mutations in single allele?  I did not see a discuss of this possibility.

2.4.1. Results:  "including a sibling of a brother ..."  Sentence not clear--please review.

3. Discussion:

"The accuracy rates ..."  Is this the same as positive predictive value?

Author Response

  1. Abstract:  "cut-off value in a continuous second blood sample"--not sure what continuous refers to.

Response:     The recall process is executed by the original Newborn Screening Center while one of the specific enzyme activity is lower than the cut-off value in the first DBS sample. The second DBS sample will be collected and referred to the same original Newborn Screening Center for the second examination. The cut-off values were set differently, for instance, the cut-off values are set at 30% of the mean activity for the first DBS and 10% of the mean activity for the second DBS. We have revised the sentence as “…in dried blood spot filter paper was lower than the cut-off value in the first and a follow-up second blood sample.” (Please see the Abstract on Page # 3).

  1. Introduction: "fallen dramatically from 4.3 to 0.2 years"--unclear how the number is so low if only 50% of infants participate in the newborn screening program.

Response:      The description in the text might be confused. From 2016-2019, about 48.5% MPS patients were diagnosed from newborn screening program in Taiwan. There were 16 confirmed infants of having MPS from the implementation of NBS for MPSs that were out of a total of 33 confirmed MPS patients from different aged groups found in Taiwan. The MPS Confirmation Center of MacKay Memorial Hospital have been assigned and cooperated with the two Newborn Screening Centers in Taiwan, in which the two NBSC handle approximately two third of newborns, about 107,000 infants per year, in Taiwan.

  • Results:  "480 cases with or without MPS or carriers"--but when we add up the case numbers it total around 480, not sure where the numbers without MPS or carriers are counted?

Response:      Apart from the 324 cases from the NBS program for MPS, the other suspected cases were from the outpatient department of MMH, from parents of confirmed and some highly suspected infants, and also from other hospitals or medical centers in Taiwan for MPS confirmation. Of the 324 cases referred from the NBS program, 20 were confirmed, of whom five, nine and six had MPS I, II, and IVA, respectively. MPS was excluded in 13 cases as no variant was found. Fifty-seven cases were defined as carriers, and the rest were categorized as highly suspected MPS cases. We have added the above descriptions into the text (Please see the RESULTS section, 2.2. The ACMG classification of gene variants from different MPS types in Taiwan, the second paragraph on Page 6).

2.3.4 Results:  "Three infants had two mutants..."--highly suspected but near normal ASB enzyme and normal GAG.  Did the authors consider cis versus trans or multiple mutations in single allele?  I did not see a discuss of this possibility.

Response:      The comment is very important for the mutant identified. In this study, we did not perform cis vs. trans analysis or analyze multiple mutations in a single allele; however, we performed Sanger sequencing for each confirmed and some of the highly suspected MPS infants and their parents for MPS I, IVA, and VI, and maternal DNA analysis was required mainly for the MPS II infants. In this way, we could precisely identify whether the variants were inherited from the father or mother. We have added the sentence in the text. (Please see the DISCUSSION section on pages 13-14.

2.4.1. Results:  "including a sibling of a brother ..."  Sentence not clear--please review.

Response:      The sentence has been revised as “a male and female sibling”. The text is revised and please see the RESULTS section on Page No. 8, the paragraph of 2.4.1. The ACMG classification of variants found in infants with confirmed and suspected MPS.

  1. Discussion:

"The accuracy rates ..."  Is this the same as positive predictive value?

Response:      The accuracy rates differ from the positive predictive value. Accurate is obtained from the True Positive # plus the True Negative # divided by the total # and multiplied by 100%, whereas the positive predictive value is obtained from the True Positive # divided by True Positive # plus False Positive # and multiplied by 100%, The Accuracy and Positive Predictive Value are usually calculated according to the following equations.

  • Positive Predictive Value =〔True Positive # / True Positive # + False Positive #〕* 100%
  • Accuracy =〔True Positive # + True Negative # / Total #〕* 100%

Reviewer 2 Report

Dear Authors,

 Thank you for this opportunity of reviewing this interesting and wonderful paper. I think  this paper would be acceptable for IJMS because of big data about neonatal mass screening for MPS in Taiwan. Also, this paper will be informative and precise about the NBS  of MPS for clinical doctors and researchers.

I have some comments and requests.

1: In this paper, all mutation description are coding variant. Could you add the description about protein variants?

2: The careful and long follow-up is needed for the highly suspected cases. How often do you check those patients?  How long do you continue the follow-up?  Those cases should be paid an attention and cared about parent's concern and anxiety. Please add the description about that in discussion.

3: In page 4, some space should be added.( The cut-off levels aresetat30%~)

Author Response

Manuscript ID: ijms-1871163
Type of manuscript: Article
Title: Updated confirmatory diagnosis for mucopolysaccharidoses in Taiwanese infants and the application of gene variants

Responses to the reviewer’s comments

We deeply appreciate the reviewer’s kind and affirmative comment and we provide our point-by-point responses to the notes of the reviewer.

Comments and Suggestions for Authors (Reviewer 2)

Dear Authors,

Thank you for this opportunity of reviewing this interesting and wonderful paper. I think this paper would be acceptable for IJMS because of big data about neonatal mass screening for MPS in Taiwan. Also, this paper will be informative and precise about the NBS of MPS for clinical doctors and researchers.

I have some comments and requests.

1: In this paper, all mutation description are coding variant. Could you add the description about protein variants?

Response:      We have added the description about protein variants in the text according to the reviewer’s comments.

2: The careful and long follow-up is needed for the highly suspected cases. How often do you check those patients?  How long do you continue the follow-up?  Those cases should be paid an attention and cared about parent's concern and anxiety. Please add the description about that in discussion.

Response:      For the confirmed and highly suspected infants, our genetic counselor will recall and arrange outpatient department visits at our hospital every 6 months for 3 years of follow-up. During long-term follow-up, our pediatric geneticists and genetic counselor pay particular attention to the parents’ concerns, and through a caring and enthusiastic approach to ease their anxiety. The above sentences have been added in the DISSCUSSION section, the last paragraph on Page 14.

3: In page 4, some space should be added.( The cut-off levels aresetat30%~)   

Response:      We have checked the whole text and revised some problems as the reviewer’s comment.